# Associations between Neighborhood Walkability, Physical Activity, and Chronic Disease in Nova Scotian Adults: An Atlantic PATH Cohort Study

**DOI:** 10.3390/ijerph17228643

**Published:** 2020-11-20

**Authors:** Melanie R. Keats, Yunsong Cui, Vanessa DeClercq, Scott A. Grandy, Ellen Sweeney, Trevor J. B. Dummer

**Affiliations:** 1Faculty of Health, School of Health and Human Performance, Dalhousie University, Halifax, NS B3H 4R2, Canada; grandy@dal.ca; 2Atlantic Partnership for Tomorrow’s Health, Faculty of Health, Dalhousie University, Halifax, NS B3H 4R2, Canada; yunsong.cui@dal.ca (Y.C.); vanessa.de.clercq@dal.ca (V.D.); ellen.sweeney@dal.ca (E.S.); 3School of Population and Public Health, University of British Columbia, Vancouver, BC V6T 1Z3, Canada; trevor.dummer@ubc.ca

**Keywords:** neighborhood walkability, physical activity, chronic disease, cohort study

## Abstract

**Background**: While neighborhood walkability has been shown to positively influence health behaviors, less is known about its impact on chronic disease. Our aim was to examine the association between walkability and self-reported physical activity in relation to chronic health conditions in an Atlantic Canadian population. **Methods**: Using data from the Atlantic Partnership for Tomorrow’s Health, a prospective cohort study, we employed both a cross-sectional and a prospective analytical approach to investigate associations of walkability and physical activity with five prevalent chronic diseases and multimorbidity. **Results**: The cross-sectional data show that participants with the lowest neighborhood walkability were more likely to have reported a pre-existing history of cancer and depression and least likely to report chronic respiratory conditions. Participants with low physical activity were more likely to have a pre-existing history of diabetes, chronic respiratory disease, and multimorbidity. Follow-up analyses showed no significant associations between walkability and chronic disease incidence. Low levels of physical activity were significantly associated with diabetes, cancer and multimorbidity. **Conclusions**: Our data provides evidence for the health protective benefits of higher levels of physical activity, and a reduction in prevalence of some chronic diseases in more walkable communities.

## 1. Introduction

As many as one-third of Canadian adults (20+ years) live with at least one of five common chronic diseases (i.e., cancer, diabetes, cardiovascular diseases, chronic respiratory diseases, and mood and/or anxiety disorders). Of these, 9% report having two or more of these chronic diseases [1]. Compared to other Canadians, Nova Scotians rank amongst the unhealthiest, with as many as 43% of the population reporting at least one and 11% reporting two or more of the five common chronic conditions [1]. With the highest proportion of Canadian seniors (65+ years; 17.2%), coupled with a high prevalence of modifiable behavioral risk factors (e.g., physical inactivity, tobacco use, alcohol consumption, unhealthy diets), there is a critical need to identify strategies to reduce the health and economic burden of chronic disease amongst Nova Scotians [2,3,4].

Given that much of the chronic disease burden has been shown to be related to a relatively small number of modifiable behavioral risk factors, improving health behaviors is often a primary target for the prevention of chronic disease and improving health outcomes [5]. While much work has been focused on individual behavioral change, it is clear that the development of chronic disease is associated with many factors, and the management and prevention of chronic disease must include a multi-level/multi-sectorial approach [6]. Specifically, behavioral change does not occur in isolation, and health behaviors are influenced by the larger socioecological context in which an individual resides. For example, there is a considerable amount of evidence showing that neighborhood design and environmental features positively influence population health by supporting engagement in healthy behaviors (i.e., increased physical activity) [7,8].

Composite measures of neighborhood walkability (i.e., design features that can promote walking and access to walkable destinations) have been consistently associated with a decreased risk of cardiometabolic diseases (e.g., obesity, hypertension, diabetes, and cardiovascular disease (CVD)) [9,10]. These protective effects are presumed to be partially attributed to increases in physical activity (i.e., higher walkability positively associated with physical activity in adults) [11]. To date, however, no Canadian study has examined the association between walkability and cancer, depression, or multimorbidity [8]. Moreover, no study has examined the association between standardized measures of walkability and health outcomes in any of the Atlantic Canadian provinces. Accordingly, the aims of this study were to (1) retrospectively examine the association between neighborhood walkability and self-reported physical activity in relation to five prevalent chronic health conditions (e.g., cancer, diabetes, cardiovascular disease, chronic respiratory disease, and depression) and multimorbidity in a sample of Nova Scotian adults, and (2) to prospectively examine incident disease risk in Nova Scotians residing in low-to-high walkability neighborhoods, while controlling for potential confounders.

## 2. Methods

### 2.1. Design

This study used data from the Atlantic Partnership for Tomorrow’s Health (PATH). Atlantic PATH is part of the larger Canadian Partnership for Tomorrow’s Health (CanPath; formerly the Canadian Partnership for Tomorrow Project), a national prospective cohort study examining the influence of genetic, environmental, and lifestyle factors on the development of chronic disease. A detailed description of the study has been previously described [12,13]. Briefly, from 2009 to 2015, 31,173 residents of the four Atlantic Canadian provinces (Nova Scotia, New Brunswick, Prince Edward Island, and Newfoundland) were invited to complete a standardized set of questionnaires designed to assess sociodemographic characteristics, health status, disease history, and lifestyle behaviors. Between 2016 and 2019, participants were invited to complete the first follow-up questionnaire aimed at updating previously collected data on sociodemographic, health, disease and lifestyle behaviors. The cross-sectional and prospective analyses in the current study are based on participants who resided in Nova Scotia at the time of data collection. The Atlantic PATH protocol was approved by the provincial and regional ethics boards, and all participants provided written informed consent prior to participation.

### 2.2. Study Area and Population

Nova Scotia is one of four Atlantic provinces located on Canada’s east coast. It is the most populated province of the Atlantic region, with a population of 977,274 [14]. Halifax is the province’s capital city and accounts for as much as 45% of the province’s populace [15]. For the purposes of aim 1, the Atlantic PATH participants included in the baseline analyses (N = 15,215) were between the ages of 35 and 69 years old, living in Nova Scotia at the time of baseline assessment (2009–2015) with non-missing walkability and physical activity data. For aim 2, the Atlantic PATH participants included in the prospective analyses (N = 6912) were living in Nova Scotia at both baseline and follow-up assessments (2016–2019), were residing in the same postal code area for at least two years, had not self-reported any of the five chronic diseases examined in aim 1, and had non-missing walkability and physical activity data. The participant data flow is shown in Figure 1.

### 2.3. Main Effects

#### 2.3.1. Walkability

Walkability was determined using the Canadian Active Living Environments (CAN-ALE) [16] dataset provided to Atlantic PATH by the Canadian Urban Environmental Health Research Consortium (CANUE). CANUE is a multidisciplinary collaboration of specialists focused on environmental exposures and population health [17]. In brief, CAN-ALE is a geography-based national metric of active living environments in Canada that can be used to facilitate direct comparisons between communities [16]. The ALE Index is a continuous measure of the favorability of the active living environment based on the summed z-scores of connectivity, dwelling density, and number of points of interest/destinations (e.g., shops, parks, businesses). Using the CAN-ALE Index, categorical walkability quintiles ranging from 1 (low walkability) to 5 (high walkability) were generated for all participants. CAN-ALE measures were linked to Atlantic PATH participant data through a six-digit residential postal code.

#### 2.3.2. Physical Activity

Physical activity was assessed using both the short and long form of the International Physical Activity Questionnaire (IPAQ) at baseline. In accordance with the IPAQ scoring protocol [18], data from both forms were used to calculate categorical (low, moderate, high) physical activity scores by sex-specific total metabolic expenditure (MET-minutes/week) tertiles [19]. Participants categorized as moderately active met any of the following three criteria: (1) engaged in ≥3 days/week of vigorous-intensity activity of at least 20 min/day; or (2) ≥5 days of moderate-intensity activity or walking at least 30 min/day; or (3) ≥5 days of any combination of walking or moderate- or vigorous-intensity activities, achieving a minimum of 6 MET-minutes/week. High active participants met either of the following two criteria: (1) engaged in vigorous-intensity activity on at least 3 days/week, accumulating 1500 MET-minutes/week; or (2) 7 days of any combination of walking, or moderate- or vigorous-intensity activities, achieving a minimum of 3000 MET-minutes/week. Those categorized as low active were not sufficiently active to meet the levels of either moderate or high activity.

### 2.4. Outcomes

#### 2.4.1. Baseline

The prevalence of chronic disease at baseline was ascertained through self-report, and included asthma, cancer, chronic obstructive pulmonary disease (COPD), chronic bronchitis, emphysema, liver cirrhosis, chronic hepatitis, myocardial infarction, stroke, hypertension, diabetes type 1/2, irritable bowel disease or irritable bowel syndrome, psoriasis or eczema, multiple sclerosis, arthritis, lupus, osteoporosis, depression, and obesity (defined by BMI ≥ 30 kg/m^2^). The five target chronic conditions (cancer—excluding skin cancer; CVD—including myocardial infarction and/or stroke; diabetes; chronic respiratory disease—including COPD, chronic bronchitis, and emphysema; and depression) were selected for analyses given their high prevalence in Canada and the Atlantic provinces. Multimorbidity was defined as ≥2 self-reported chronic conditions.

#### 2.4.2. Follow-Up

Target chronic conditions were defined as self-reported, new (i.e., not reported at baseline) conditions that were revealed at follow-up, and included cancer (excluding skin cancer), CVD (myocardial infarction or stroke), diabetes, chronic respiratory disease, depression, and multi-morbidity (as defined above).

### 2.5. Covariables

Baseline characteristics, including sex, age, daily servings of fruit and vegetables, alcohol consumption, and smoking status were ascertained from Atlantic PATH participants’ baseline data.

#### 2.5.1. Daily Fruit and Vegetable Consumption

Total daily servings of fruit and vegetables were assessed by the following two questions: (1) In a typical day, how many servings of vegetables do you eat? One serving is about ½ cup or 125 mL of fresh, frozen, canned or cooked vegetables and (2) In a typical day, how many servings of fruit do you eat? One serving is about ½ cup or 125 mL of fresh, frozen, or canned fruit. For the present study, the number of daily servings of fruits and vegetables was summed. The combined value was expressed as the overall number of daily servings of fruit and vegetables. Adequate fruit and vegetable intake were defined as consuming at least 5 servings of combined daily servings of fruits and vegetables [20].

#### 2.5.2. Alcohol Consumption

Participants were asked to indicate whether they had ever consumed alcohol (yes/no). If yes, they were asked to indicate the average frequency of alcohol consumption over the last year. Based on participant response they were classified as abstainer (never consumed alcohol), occasional drinker (>0 to ≤2–3 times/month), regular drinker (≥1 time/week to ≤2–3 times/week), or habitual drinker (≥4–5 times/week) [20].

#### 2.5.3. Smoking

Participants were first asked if they had smoked at least 100 cigarettes in their lifetime. If yes, they were asked to indicate at what age they smoked their first whole cigarette, what their smoking behavior was at present, at what age they began daily smoking, how many cigarettes per day they smoke now (or did when a daily smoker), and for how many years they were a daily smoker. Participants were then categorized as non-smoker (never having smoked 100 cigarettes in their life or other tobacco products on a regular basis for at least six months), former smoker (reported having at least 100 cigarettes in their lifetime but no tobacco use in the previous 30 days), and current smoker (those who reported smoking at least 100 cigarettes in their lifetime and smoked within the past 30 days) [20].

#### 2.5.4. Material and Social Deprivation Score

Material and social deprivation scores were obtained through partnership with CANUE [17]. The domains related to the material dimension reflect the proportion of people with no high school diploma, average household income, and employment rate. The domains related to the social dimension are the proportion of individuals who are separated, divorced, or widowed, the proportion of single-parent families, and the proportion of persons living alone [21,22]. Material and social deprivation quintiles were generated by ranking participants on material and social deprivation scores at baseline and follow-up. Data merging with Atlantic PATH was facilitated through six-digit residential postal code linkage.

#### 2.5.5. Rurality

Rurality was ascertained from the CANUE data set. The geographic area variable was categorized as follows: (1) a large census metropolitan area (CMA; includes the three largest metropolitan areas in Canada—Montreal, Toronto, and Vancouver); (2) other CMA (defined as a core population greater than 100,000); (3) census agglomeration (CA; defined as having a core population between 10,000 and 100,000); and (4) rural (defined as all areas inside the CMA or CA that are not core or fringe).

### 2.6. Analyses

We assessed the baseline characteristics of study participants by walkability quintile, mean and standard deviation for continuous variables, and frequency and percentage for dichotomous/nominal variables (Table 1). We then applied adjusted (sex, age, material and social deprivation, geographical area, and self-reported physical activity, smoking behaviors, and alcohol and fruit and vegetable consumption) logistic regression models to investigate the associations of walkability and physical activity with each chronic disease and multimorbidity outcome. Walkability and physical activity were estimated simultaneously in the adjusted model. High walkability and high physical activity were set as reference levels (Table 2). We then repeated all baseline analyses on the follow-up population; note that only those participants who did not report any of the five target chronic conditions were included in the final analyses (Table 3 and Table 4). Data management and analyses were performed with SAS statistical package version 9.4 (SAS Institute, Cary, North Carolina) and R package version 3.6.3 [23]. 

## 3. Results

### 3.1. Cross-Sectional Study Findings

Participant characteristics are presented by walking quintiles in Table 1. Participants with a high level of walkability were more likely to be more socially deprived, but less likely to be materially deprived. Rural participants were less likely to live in a highly walkable neighborhood and high physical activity levels were found to decrease with higher neighborhood walkability.

Table 2 displays the results for the associations between walkability and physical activity with selected health conditions and multimorbidity. Low levels of physical activity were significantly associated with higher odds of diabetes (OR 1.78, 95% CI 1.5–2.12, *p* < 0.0001). No significant association was found for low walkability (OR 1.11, 95% CI 0.83–1.47, *p* = 0.493).

With the exception of the second quintile, low to moderate levels of walkability were significantly associated with higher odds of cancer. Of note, higher walkability levels (quintile 4) were also shown to relate to significantly higher odds of cancer (OR 2.14, 95% CI 1.46–3.13, *p* < 0.0001, respectively). Physical activity was not significantly associated with higher odds of cancer.

Higher odds of depression were found for low levels of both walkability (OR 2.08, 95% CI 1.47–2.95, *p* < 0.0001) and physical activity (OR 1.21, 95% CI 0.99–1.48, *p* = 0.0579). Similar to the association with cancer, higher walkability (quintile 4) was shown to relate to significantly higher odds of depression (OR 1.75, 95% CI 1.3–2.36, *p* = 0.0002).

No significant associations were found for either low walkability or physical activity and CVD.

Compared to those in highly walkable neighborhoods, residents in the least walkable neighborhoods were significantly less likely to have chronic respiratory disease (OR = 0.62, 95% CI 0.43–0.89, *p* = 0.0102). Participants with low levels of physical activity were significantly more likely to have chronic respiratory disease (OR = 1.27, 95% CI 1.03–1.56, *p* = 0.0244).

Low physical activity, but not walkability, was significantly associated with multimorbidity (OR 1.35, 95% CI 1.24–1.47, *p* < 0.0001). As noted with both cancer and depression, higher walkability (quintile 4) was shown to relate to significantly higher odds of multimorbidity (OR 1.14, 95% CI 1.02–1.28, *p* = 0.0247 respectively).

### 3.2. Prospective Study Findings

Table 3 shows the sociodemographic and health characteristics of the 6912 Nova Scotia participants who completed the follow-up questionnaire, who were free from the five target chronic conditions (i.e., cancer, CVD, diabetes, depression, and chronic respiratory disease) and were residing within the same postal code for at least for 2 years. The findings mirror those of the baseline cross-sectional findings where participants with the highest level of walkability were most likely to be socially deprived, but least likely to be materially deprived. Rural residents were again shown to have the least walkable neighborhoods. No clear trend was noted for physical activity and walkability at follow-up.

Table 4 displays the results for the associations between walkability and baseline physical activity with selected health conditions and multimorbidity at follow-up. Low levels of baseline physical activity were significantly associated with higher odds of diabetes (OR 1.81, 95% CI 1.15–2.85, *p* = 0.010). No significant associations were found for walkability.

Low levels of baseline physical activity were associated with higher odds of cancer (OR 1.41, 95% CI 1.09–1.83, *p* = 0.0092). No significant associations were found with lower levels of walkability and odds of cancer.

No significant associations between either low walkability or levels of baseline physical activity and odds of depression were found.

No significant association was found between walkability and CVD. Moderate levels of baseline physical activity were shown to be significantly associated with decreased odds of CVD (OR 0.40, 95% CI 0.18–0.87, *p* = 0.0213).

There were no significant findings for the relationship between walkability or physical activity and chronic respiratory disease.

Low levels of baseline physical activity, but not low walkability, were significantly associated with higher odds of multimorbidity (OR 1.26, 95% CI 1.11–1.42, *p* = 0.0002).

## 4. Discussion

Several modifiable risk behaviors have been associated with the physical environment. For example, a growing body of research has shown that the physical environment plays an important role in supporting an active lifestyle through the collective availability of activity-friendly neighborhood characteristics (i.e., walkability) across several cultural contexts [11,24,25,26]. As physical activity has been well-established to reduce chronic disease risk [27], residents of highly walkable communities should, by extension, present with better health outcomes [28].

While previous studies have observed a positive association between neighborhood walkability and physical activity [11,25,29,30], these findings are inconsistent with our own. We did not find a positive linear trend in either low or moderate levels of physical activity and walkability at baseline. A notable finding was the downward linear trend in high levels of physical activity with higher walkability at baseline. No clear association was found between any level of physical activity and walkability at follow-up. Although it is not clear why this relationship/lack of relationship emerged, it is well-established that physical activity behaviors are influenced by a complex interplay of intra- (e.g., social support) and inter-personal (e.g., knowledge, motivation), cultural, economic, and environmental factors. For example, residential self-selection has been shown to influence the impact of the built environment on an individual’s or group’s physical activity behaviors [30,31]. Those more inclined to walk for transportation (i.e., intra-personal influence) may seek out and reside in neighborhoods which provide greater access to amenities within walking distance. Those who are less inclined to walk for any purpose are not likely to choose a neighborhood based on perceived walkability [30]. Research has also shown that some neighborhood characteristics are more important than others with respect to supporting different types of walking (i.e., walking for transportation vs. walking for recreation) [32]. These findings are consistent with the work of others who have shown that walkability is positively correlated with walking for transport but shows no or negative associations with recreational physical activity [28]. Thus, the relationship between neighborhood walkability and physical activity can be influenced by the characteristics of the environment itself, an individual’s preference for physical activity, or both [31].

The findings from the cross-sectional data are partially consistent with others and suggest that compared to individuals in the highest walkability category, participants with the lowest neighborhood walkability were more likely to have reported a pre-existing history of cancer and depression [8]. Those residing in low walkable neighborhoods were less likely to report a pre-existing history of a chronic respiratory disease. While we are only able to speculate, the decreased prevalence of chronic respiratory conditions may be partially attributable to the reduced adverse impacts of exposure to air pollution, which are typically found in more walkable urban centers [33,34]. Notably, participants residing in the higher walkability categories (quintiles 3 and/or 4) were more likely to report having cancer, depression and multimorbidity. These incongruent findings suggest that while activity-friendly neighborhoods may support health-promoting behaviors (e.g., physical activity), the benefits are limited to those who are willing to engage in the behavior(s).

Compared to those with the highest level of physical activity, participants with low physical activity were more likely to have a pre-existing history of diabetes, chronic respiratory disease, and multimorbidity. These findings highlight the importance of physical activity in both the primary and secondary prevention of chronic disease [27].

At follow-up, no significant associations were found between neighborhood walkability and incident chronic disease or multimorbidity. As walkability was not linearly associated with physical activity in our models, our findings reinforce the notion that if you build it, they may or may not come. Consistent with the work of others, the protective health benefits of neighborhood walkability appear to be at least partially mediated by physical activity levels [10].

Overall, the findings from the prospective study suggest a stronger relationship between physical activity and chronic disease. Specifically, participants with low activity were significantly more likely to be diagnosed with diabetes and cancer. Less active participants were also significantly more likely to have multiple chronic conditions. These findings are again in alignment with the growing body of evidence which demonstrates a decreased risk of chronic illness with increasing levels of physical activity [27].

### Strengths and Limitations

To the best of our knowledge, this is the first Canadian study to examine the association between walkability and cancer, depression and multimorbidity, and is the first study to consider the association between standardized measures of walkability and health outcomes in Atlantic Canada. This study utilized a large sample of Nova Scotia participants and accounted for sociodemographic moderators (e.g., age, sex, material and social deprivation, rurality), as well as controlling for confounding factors (e.g., smoking history, alcohol consumption, physical activity). A significant strength of the study was our ability to include both prevalent and incident cases of five common chronic diseases and multimorbidities by utilizing both baseline and follow-up data.

We do acknowledge the limitations of our study, such as being limited by self-reported physical activity and chronic disease incidence/prevalence data (i.e., data subject to social biases and/or personal knowledge and willingness to report). We were also not able to assess disease severity, which may impact activity levels. Further, the cross-sectional design is limited by possible reverse causality.

Finally, there are limitations with the self-reported IPAQ data, though it is a validated and widely used measure of physical activity [35]. As we combined data from both the long- and short-form of the IPAQ measure, we used an aggregate measure of physical activity, which was categorized as low, moderate, and high. In doing so, we were not able to examine domain-specific associations between neighborhood walkability and physical activity levels. Further, as data on physical activity were not collected at the first follow-up, we were constrained to a single, self-reported measure of baseline physical activity across all analyses. Future research will include new reports of physical activity and health outcomes collected as part of the prospective nature of Atlantic PATH. We will also consider regional differences in walkability, physical activity and chronic disease by expanding this work to the other regional CanPath cohorts across Canada.

## 5. Conclusions

In sum, we did not find a positive association between walkability and chronic disease incidence in the current study. However, our data provide evidence for the health protective benefits of higher levels of physical activity, and a reduction in prevalence of some chronic diseases in more walkable communities. Thus, highly walkable neighborhoods may confer health benefits for those residents willing and able to engage in physical activity. An additional study is needed in order to better understand the interactional effects of both individual-level influences and the broader social and cultural climate on population-level behavior change. Moreover, there is a need for a culture shift whereby unhealthy behaviors, such as physical inactivity, are denormalized [36]. Environments which are activity-friendly and improve access to recreational resources, such as parks, playgrounds, and green space, can reduce perceived barriers to activity and can enforce activity as a social and cultural norm [37].

## Figures and Tables

**Figure 1 ijerph-17-08643-f001:**
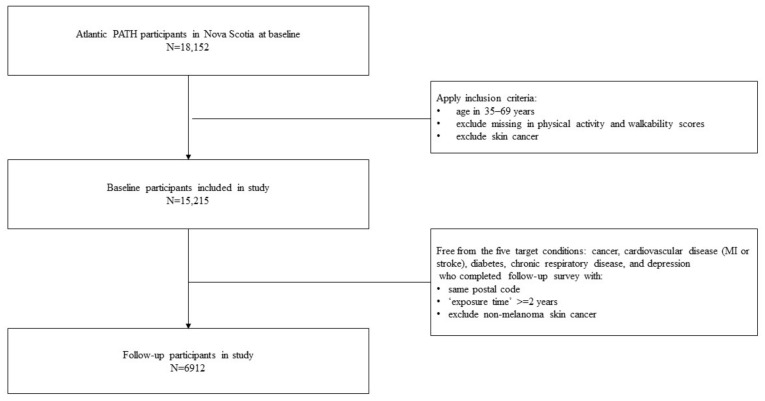
Participant data flow.

**Table 1 ijerph-17-08643-t001:** Sociodemographic and health-related participant characteristics by walking quintile.

Variables	Walkability Quintile (Based on ale_06 ^a^)					
	Q1 (Least)	Q2	Q3	Q4	Q5 (Most)	
	*n* = 3310	*n* = 2798	*n* = 3097	*n* = 2983	*n* = 3027	*p*-Value ^f^
Sex, N (%)						0.7224
Male	977 (29.5)	848 (30.3)	932 (30.1)	925 (31)	932 (30.8)	
Female	2333 (70.5)	1950 (69.7)	2165 (69.9)	2058 (69)	2095 (69.2)	
Age (SD)	53.0 (9.2)	53.7 (9.0)	53.0 (9.0)	53.2 (9.2)	53.8 (9.0)	0.0004
Age group, N (%)						0.0008
35–49	1192 (36)	928 (33.2)	1100 (35.5)	1044 (35)	955 (31.5)	
50–59	1152 (34.8)	970 (34.7)	1126 (36.4)	1051 (35.2)	1120 (37)	
60–69	966 (29.2)	900 (32.2)	871 (28.1)	888 (29.8)	952 (31.5)	
Rurality ^b^, N (%)						<0.0001
No	1453 (43.9)	2021 (72.2)	2423 (78.2)	2643 (88.6)	2964 (97.9)	
Yes	1627 (49.2)	713 (25.5)	565 (18.2)	272 (9.1)	45 (1.5)	
Unknown	230 (6.9)	64 (2.3)	109 (3.5)	68 (2.3)	18 (0.6)	
Material deprivation ^c^						<0.0001
(quintile), N (%)						
Q1 (Least deprived)	122 (3.7)	473 (16.9)	562 (18.1)	1056 (35.4)	1387 (45.8)	
Q2	397 (12)	503 (18)	675 (21.8)	535 (17.9)	632 (20.9)	
Q3	733 (22.1)	699 (25)	594 (19.2)	490 (16.4)	336 (11.1)	
Q4	1062 (32.1)	609 (21.8)	609 (19.7)	375 (12.6)	423 (14)	
Q5 (Most deprived)	704 (21.3)	381 (13.6)	509 (16.4)	319 (10.7)	157 (5.2)	
Unknown	292 (8.8)	133 (4.8)	148 (4.8)	208 (7)	92 (3)	
Social deprivation ^d^						
(quintile), N (%)						<0.0001
Q1 (Least deprived)	751 (22.7)	930 (33.2)	689 (22.2)	497 (16.7)	179 (5.9)	
Q2	1382 (41.8)	776 (27.7)	699 (22.6)	568 (19)	368 (12.2)	
Q3	538 (16.3)	521 (18.6)	479 (15.5)	203 (6.8)	251 (8.3)	
Q4	308 (9.3)	422 (15.1)	742 (24)	887 (29.7)	900 (29.7)	
Q5 (Most deprived)	39 (1.2)	16 (0.6)	340 (11)	620 (20.8)	1237 (40.9)	
Unknown	292 (8.8)	133 (4.8)	148 (4.8)	208 (7)	92 (3)	
Physical activity ^e^						<0.0001
Low	1080 (32.6)	863 (30.8)	1037 (33.5)	1082 (36.3)	1011 (33.4)	
Moderate	1074 (32.4)	951 (34)	1019 (32.9)	946 (31.7)	1082 (35.7)	
High	1156 (34.9)	984 (35.2)	1041 (33.6)	955 (32)	934 (30.9)	
Smoking status						0.0072
Never smoked	1599 (48.3)	1388 (49.6)	1514 (48.9)	1508 (50.6)	1573 (52)	
Former smoker	1306 (39.5)	1146 (41)	1249 (40.3)	1175 (39.4)	1160 (38.3)	
Current smoker	369 (11.1)	246 (8.8)	304 (9.8)	280 (9.4)	277 (9.2)	
Unknown	36 (1.1)	18 (0.6)	30 (1)	20 (0.7)	17 (0.6)	
Daily fruits/vegetable						0.2942
intake						
≤4	1680 (50.8)	1445 (51.6)	1600 (51.7)	1536 (51.5)	1639 (54.1)	
≥5	1551 (46.9)	1296 (46.3)	1421 (45.9)	1381 (46.3)	1325 (43.8)	
Unknown	79 (2.4)	57 (2)	76 (2.5)	66 (2.2)	63 (2.1)	
Alcohol consumption						<0.0001
Abstainer	231 (7)	182 (6.5)	191 (6.2)	172 (5.8)	186 (6.1)	
Occasional drinker	994 (30)	796 (28.4)	959 (31)	821 (27.5)	745 (24.6)	
Regular	1389 (42)	1225 (43.8)	1379 (44.5)	1390 (46.6)	1306 (43.1)	
Habitual drinker	522 (15.8)	486 (17.4)	446 (14.4)	491 (16.5)	676 (22.3)	
Unknown	174 (5.3)	109 (3.9)	122 (3.9)	109 (3.7)	114 (3.8)	

Note: Q1—Quintile 1 (low); Q2—Quintile 2; Q3—Quintile 3; Q4—Quintile 4; Q5—Quintile 5 (high); ^a^ Active Living Environment (ALE) Index—sum of all z-scores (Atlantic Partnership for Tomorrow’s Health (PATH) Cohort linked with Canadian Active Living Environments (Can-ALE) walkability data). The following variables were linked to Atlantic PATH data from the Canadian Urban Environmental Health Research Consortium. ^b^ Based on MSD_06-Geographical area—Large census metropolitan area, other census metropolitan area, census agglomeration, and rural; ^c^ Based on MSD_08-Deprivation index—Material factor score; ^d^ Based on MSD_09-Deprivation index—Social factor score; ^e^ International Physical Activity Questionnaire (IPAQ) data from both long and short forms were used to calculate categorical (low, moderate, high) physical activity scores by sex-specific total metabolic expenditure (MET-minutes/week); ^f^ F-test for continuous variables and Chi-Square test for categorical variables.

**Table 2 ijerph-17-08643-t002:** Associations of walkability and physical activity with selected chronic conditions and multi-morbidity at baseline.

Main Effects	Cases	Adjusted Model ^a^	
		OR (95% CI)	*p*-Value
Diabetes			
Walkability quintile			
Q1 (Least)	208	1.11 (0.83,1.47)	0.493
Q2	146	0.98 (0.74,1.3)	0.8826
Q3	172	1.06 (0.82,1.36)	0.6681
Q4	168	1.15 (0.9,1.46)	0.2557
Q5 (Most)	151	1 (Ref)	-
Physical activity			
Low	365	1.78 (1.5,2.12)	<0.0001
Moderate	255	1.19 (0.99,1.43)	0.0669
High	225	1 (Ref)	-
Cancer (excluding skin cancer)			
Walkability quintile			
Q1 (Least)	165	2.96 (1.94,4.54)	<0.0001
Q2	53	1.27 (0.81,2.01)	0.3009
Q3	84	1.8 (1.2,2.69)	0.0045
Q4	91	2.14 (1.46,3.13)	<0.0001
Q5 (Most)	43	1 (Ref)	-
Physical activity			
Low	132	0.85 (0.67,1.08)	0.1738
Moderate	143	0.93 (0.73,1.17)	0.5119
High	161	1 (Ref)	-
Depression			
Walkability quintile			
Q1 (Least)	220	2.08 (1.47,2.95)	<0.0001
Q2	56	0.77 (0.52,1.15)	0.1972
Q3	110	1.28 (0.92,1.78)	0.135
Q4	135	1.75 (1.3,2.36)	0.0002
Q5 (Most)	76	1 (Ref)	-
Physical activity			
Low	234	1.21 (0.99,1.48)	0.0579
Moderate	173	0.93 (0.75,1.15)	0.4902
High	190	1 (Ref)	-
CVD (MI or Stroke)			
Walkability quintile			
Q1 (Least)	98	1.27 (0.85,1.90)	0.2412
Q2	70	1.08 (0.73,1.59)	0.7127
Q3	77	1.05 (0.74,1.50)	0.7900
Q4	74	1.04 (0.74,1.45)	0.8345
Q5 (Most)	81	1 (Ref)	-
Physical activity			
Low	149	1.26 (0.98,1.61)	0.0667
Moderate	124	1.01 (0.78,1.30)	0.9487
High	127	1 (Ref)	-
Chronic respiratory conditions ^b^			
Walkability quintile			
Q1 (Least)	85	0.62 (0.43,0.89)	0.0102
Q2	106	1.01 (0.73,1.4)	0.9504
Q3	107	0.89 (0.66,1.2)	0.4476
Q4	103	0.91 (0.68,1.2)	0.4877
Q5 (Most)	121	1 (Ref)	-
Physical activity			
Low	209	1.27 (1.03,1.56)	0.0244
Moderate	140	0.82 (0.66,1.03)	0.0958
High	173	1 (Ref)	-
Multi-morbidity ^c^			
Walkability quintile			
Q1 (Least)	1261	1.01 (0.88,1.17)	0.8406
Q2	1008	0.99 (0.87,1.13)	0.8567
Q3	1175	1.08 (0.96,1.22)	0.2132
Q4	1137	1.14 (1.02,1.28)	0.0247
Q5 (Most)	1074	1 (Ref)	-
Physical activity			
Low	2084	1.35 (1.24,1.47)	<0.0001
Moderate	1762	1 (0.92,1.08)	0.9448
High	1809	1 (Ref)	-

Notes: Q1—Quintile 1 (low); Q2—Quintile 2; Q3—Quintile 3; Q4—Quintile 4; Q5—Quintile 5 (high); the interaction term for physical activity and Active Living Environment (walkability) for each of the outcomes was tested in the model and were not significant at the 0.05 level; ^a^ adjusted for sex, age, rurality, socioeconomic status (both material/social deprivation scores quintile), alcohol consumption, daily servings of fruit and vegetables, smoking status, and physical activity/walkability where appropriate; ^b^ chronic obstructive pulmonary disease, emphysema, and chronic bronchitis; ^c^ number of all chronic conditions ≥2.

**Table 3 ijerph-17-08643-t003:** Sociodemographic and health-related participant characteristics categorized by walking quintile at follow-up.

Variables	Walkability Quintile (Based on ale_06 ^a^)					
	Q1 (Least)	Q2	Q3	Q4	Q5 (Most)	
	*n* = 1505	*n* = 1268	*n* = 1375	*n* = 1388	*n* = 1374	*p*-Value ^f^
Sex, N (%)						0.0352
Male	399 (26.5)	371 (29.3)	413 (30)	442 (31.8)	405 (29.4)	
Female	1106 (73.5)	897 (70.7)	962 (70)	946 (68.2)	971 (70.6)	
Age (SD)	53.3 (8.8)	53.4 (8.7)	53.5 (8.7)	53.3 (8.8)	54.0 (8.7)	
Age group, N (%)						0.4170
35–49	512 (34)	419 (33)	455 (33.1)	480 (34.6)	417 (30.3)	
50–59	567 (37.7)	465 (36.7)	519 (37.7)	500 (36)	534 (38.8)	
60–69	426 (28.3)	384 (30.3)	401 (29.2)	408 (29.4)	425 (30.9)	
Rurality ^b^, N (%)						<0.0001
No	649 (43.1)	910 (71.8)	1050 (76.4)	1219 (87.8)	1363 (99.1)	
Yes	730 (48.5)	326 (25.7)	262 (19.1)	143 (10.3)	6 (0.4)	
Unknown	126 (8.4)	32 (2.5)	63 (4.6)	26 (1.9)	7 (0.5)	
Material deprivation ^c^						
(quintile), N (%)						<0.0001
Q1 (Least deprived)	64 (4.3)	232 (18.3)	263 (19.1)	507 (36.5)	669 (48.6)	
Q2	187 (12.4)	211 (16.6)	299 (21.7)	260 (18.7)	304 (22.1)	
Q3	337 (22.4)	331 (26.1)	259 (18.8)	275 (19.8)	120 (8.7)	
Q4	471 (31.3)	243 (19.2)	255 (18.5)	147 (10.6)	182 (13.2)	
Q5 (Most deprived)	293 (19.5)	194 (15.3)	224 (16.3)	124 (8.9)	59 (4.3)	
Unknown	153 (10.2)	57 (4.5)	75 (5.5)	75 (5.4)	42 (3.1)	
Social deprivation ^d^						
(quintile), N (%)						<0.0001
Q1 (Least deprived)	329 (21.9)	432 (34.1)	342 (24.9)	249 (17.9)	90 (6.5)	
Q2	629 (41.8)	381 (30)	298 (21.7)	311 (22.4)	192 (14)	
Q3	241 (16)	221 (17.4)	228 (16.6)	133 (9.6)	130 (9.4)	
Q4	125 (8.3)	127 (10)	242 (17.6)	280 (20.2)	295 (21.4)	
Q5 (Most deprived)	28 (1.9)	50 (3.9)	190 (13.8)	340 (24.5)	627 (45.6)	
Unknown	153 (10.2)	57 (4.5)	75 (5.5)	75 (5.4)	42 (3.1)	
Physical activity ^e^						0.0889
Low	496 (33)	400 (31.5)	469 (34.1)	494 (35.6)	445 (32.3)	
Moderate	497 (33)	414 (32.6)	439 (31.9)	459 (33.1)	496 (36)	
High	512 (34)	454 (35.8)	467 (34)	435 (31.3)	435 (31.6)	
Smoking status						0.0014
Never smoked	802 (53.3)	649 (51.2)	704 (51.2)	777 (56)	776 (56.4)	
Former smoker	558 (37.1)	531 (41.9)	559 (40.7)	515 (37.1)	506 (36.8)	
Current smoker	126 (8.4)	82 (6.5)	105 (7.6)	90 (6.5)	87 (6.3)	
Unknown	19 (1.3)	6 (0.5)	7 (0.5)	6 (0.4)	7 (0.5)	
Daily fruits/vegetables intake						0.0357
≤4	726 (48.2)	634 (50)	669 (48.7)	661 (47.6)	735 (53.4)	
≥5	748 (49.7)	612 (48.3)	677 (49.2)	710 (51.2)	621 (45.1)	
Unknown	31 (2.1)	22 (1.7)	29 (2.1)	17 (1.2)	20 (1.5)	
Alcohol consumption						<0.0001
Abstainer	88 (5.8)	69 (5.4)	62 (4.5)	69 (5)	66 (4.8)	
Occasional drinker	416 (27.6)	340 (26.8)	397 (28.9)	362 (26.1)	272 (19.8)	
Regular	679 (45.1)	586 (46.2)	645 (46.9)	679 (48.9)	639 (46.4)	
Habitual drinker	242 (16.1)	233 (18.4)	230 (16.7)	234 (16.9)	354 (25.7)	
Unknown	80 (5.3)	40 (3.2)	41 (3)	44 (3.2)	45 (3.3)	

Note: Q1—Quintile 1 (low); Q2—Quintile 2; Q3—Quintile 3; Q4—Quintile 4; Q5—Quintile 5 (high); ^a^ Active Living Environment (ALE) Index—sum of all z-scores (Atlantic Partnership for Tomorrow’s Health (PATH) Cohort linked with Canadian Active Living Environments (Can-ALE) walkability data); The following variables were linked to the Atlantic PATH data from the Canadian Urban Environmental Health Research Consortium; ^b^ Based on MSD_06-Geographical area—Large census metropolitan area, other census metropolitan area, census agglomeration, and rural; ^c^ Based on MSD_08-Deprivation index—material factor score; ^d^ Based on MSD_09-Deprivation index—Social factor score; ^e^ International Physical Activity Questionnaire (IPAQ) data from both long and short forms were used to calculate categorical (low, moderate, high) physical activity scores by sex-specific total metabolic expenditure (MET-minutes/week); ^f^ F-test for continuous variables and Chi-Square test for categorical variables.

**Table 4 ijerph-17-08643-t004:** Associations of walkability and physical activity with selected chronic conditions and multi-morbidity at follow-up.

		Adjusted Model ^a^	
Main Effects	Cases	OR (95% CI)	*p*-Value
Diabetes			
Walkability quintile			
Q1 (Least)	33	1.45 (0.68,3.09)	0.3323
Q2	25	1.3 (0.63,2.7)	0.4743
Q3	28	1.4 (0.71,2.74)	0.3331
Q4	23	1.24 (0.64,2.39)	0.5288
Q5 (Most)	17	1 (Ref)	-
Physical activity			
Low	52	1.81 (1.15,2.85)	0.010
Moderate	43	1.45 (0.91,2.31)	0.1208
High	31	1 (Ref)	-
Cancer			
(non-melanoma skin cancer excluded)			
Walkability quintile			
Q1 (Least)	76	1.01 (0.66,1.54)	0.9669
Q2	64	0.88 (0.59,1.31)	0.5304
Q3	77	0.96 (0.67,1.38)	0.817
Q4	80	1.01 (0.72,1.41)	0.9654
Q5 (Most)	81	1 (Ref)	-
Physical activity			
Low	142	1.41 (1.09,1.83)	0.0092
Moderate	126	1.18 (0.9,1.54)	0.2232
High	110	1 (Ref)	-
Depression			
Walkability quintile			
Q1 (Least)	6	1.11 (0.31,4.02)	0.8683
Q2	4	0.58 (0.15,2.22)	0.4241
Q3	8	1 (0.33,3.02)	0.9975
Q4	7	0.81 (0.28,2.33)	0.6988
Q5 (Most)	9	1 (Ref)	-
Physical activity			
Low	13	1.29 (0.55,3.06)	0.5574
Moderate	12	1.24 (0.52,2.96)	0.6336
High	9	1 (Ref)	-
CVD (MI or Stroke)			
Walkability quintile			
Q1 (Least)	5	0.52 (0.14,1.97)	0.3355
Q2	12	1.63 (0.57,4.65)	0.3635
Q3	14	1.61 (0.63,4.16)	0.3232
Q4	8	0.92 (0.34,2.48)	0.866
Q5 (Most)	9	1 (Ref)	-
Physical activity			
Low	16	0.74 (0.39,1.42)	0.3656
Moderate	9	0.4 (0.18,0.87)	0.0213
High	23	1 (Ref)	-
Chronic respiratory disease ^b^			
Walkability quintile			
Q1 (Least)	11	1.7 (0.51,5.7)	0.3863
Q2	9	1.8 (0.58,5.64)	0.3101
Q3	11	1.59 (0.56,4.51)	0.3866
Q4	10	1.63 (0.6,4.44)	0.3429
Q5 (Most)	7	1 (Ref)	-
Physical activity			
Low	22	1.51 (0.79,2.89)	0.2175
Moderate	10	0.63 (0.29,1.4)	0.2613
High	16	1 (Ref)	-
Multi-morbidity ^c^			
Walkability quintile			
Q1 (Least)	554	0.95 (0.78,1.17)	0.6413
Q2	493	1.08 (0.89,1.31)	0.417
Q3	551	1.12 (0.94,1.34)	0.2035
Q4	548	1.17 (0.99,1.38)	0.0642
Q5 (Most)	492	1 (Ref)	-
Physical activity			
Low	958	1.26 (1.11,1.42)	0.0002
Moderate	817	0.94 (0.83,1.06)	0.3143
High	863	1 (Ref)	-

Notes: Q1—Quintile 1 (low); Q2—Quintile 2; Q3—Quintile 3; Q4—Quintile 4; Q5—Quintile 5 (high); the interaction term for physical activity and the Active Living Environment (walkability) for each of the outcomes was tested in the model, and were not significant at the 0.05 level; ^a^ adjusted for sex, age, rurality, socioeconomic status (both material/social deprivation scores quintile), alcohol consumption, daily servings of fruit and vegetables, smoking status, and physical activity/walkability where appropriate; ^b^ chronic obstructive pulmonary disease, emphysema, and chronic bronchitis; ^c^ number of all chronic conditions ≥2.

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
