# Peer review of "Associations between Neighborhood Walkability, Physical Activity, and Chronic Disease in Nova Scotian Adults: An Atlantic PATH Cohort Study"

_ijerph, 2020, doi:10.3390/ijerph17228643_

Round 1

Reviewer 1 Report

This study investigated associations of walkability and physical activity with five prevalent chronic diseases and multimorbidity using both a cross-sectional and prospective analytical approach. The significance of this study lies in the prospective analysis, considering the large number of cross-sectional studies and the problem of residential self-selection, as the authors pointed out. I hope that it will be revised so that the significance of the study is clearly presented.

Major points

L.62

As many as three aims make this article redundant. Especially in relation to the second aim, the unadjusted models are less necessary, obscure the results section, and are not fully discussed in the discussion section.

L105

Participants are nested in postal code areas. Why didn't you use multi-level models?

L.216

The significance level should be 5% due to the large sample size. The results section is confusing as it also mentions non-significant results. (For example, L.222 "marginally significant" or L.274 "Although not significant")

L.372-386

These two paragraphs are the most important in the discussion section, but the points are unclear because they are written with a mixture of significant and non-significant results. The result that walkability was not significantly related to five prevalent chronic diseases or multimorbidity in the prospective analysis is the greatest finding of this study.

L.410

The Conclusions section should be written based on the analysis results. Parks, playgrounds, and green spaces are irrelevant.

Minor points

L.92

Is "N = 7,158" correct? What is the relationship with "N = 6,912" in Fig. 1?

L.187

It is unknown whether walkability and physical activity were estimated simultaneously in one model or in different models. You had better to explain in the text instead of table notes. I don't think uni-variate models are needed.

L.331

The fact that walking for transportation and walking for recreation could not be distinguished should be written in limitation section. Long discussion is unnecessary here.

L. 340

The discussion of material and social deprivation is not the main concern of your study. I think that a long discussion is unnecessary.

L.372

What do you mean by the term "mixed effects"? Usually refers to a statistical model containing both fixed effects and random effects.

Author Response

Reviewer #1

Comments and Suggestions for Authors

This study investigated associations of walkability and physical activity with five prevalent chronic diseases and multimorbidity using both a cross-sectional and prospective analytical approach. The significance of this study lies in the prospective analysis, considering the large number of cross-sectional studies and the problem of residential self-selection, as the authors pointed out. I hope that it will be revised so that the significance of the study is clearly presented.

Major points

L.62

As many as three aims make this article redundant. Especially in relation to the second aim, the unadjusted models are less necessary, obscure the results section, and are not fully discussed in the discussion section.

We have removed the unadjusted models from Tables 2 and 4 and the corresponding Results and Discussion sections.

L105

Participants are nested in postal code areas. Why didn't you use multi-level models?

While a multi-level model was considered, we opted not to adopt  the approach  because we only included a single geographical risk factor (i.e. walkability) at a time in the model (i.e. potential confounders were individual level factors).

 L.216

The significance level should be 5% due to the large sample size. The results section is confusing as it also mentions non-significant results. (For example, L.222 "marginally significant" or L.274 "Although not significant")

We have revised the manuscript to only present and discuss the statistically significant findings.

L.372-386

These two paragraphs are the most important in the discussion section, but the points are unclear because they are written with a mixture of significant and non-significant results. The result that walkability was not significantly related to five prevalent chronic diseases or multimorbidity in the prospective analysis is the greatest finding of this study.

We have removed the non-significant and marginally significant findings from the Results and Discussion sections. We highlight that no significant associations were found between walkability and incident chronic disease.

L.410

The Conclusions section should be written based on the analysis results. Parks, playgrounds, and green spaces are irrelevant.

We have revised the concluding statement in an effort to improve clarity. As both neighborhood walkability (proximity to points of interest – such as parks) and physical activity are influenced by proximity and access to supportive resources, we have concluded that access and use of these resources will be instrumental in denormalizing physical inactivity.

Minor points

L.92

Is "N = 7,158" correct? What is the relationship with "N = 6,912" in Fig. 1?

We have revised the text in L92 to reflect the correct sample size (6,912) as illustrated in Figure 1.

L187

It is unknown whether walkability and physical activity were estimated simultaneously in one model or in different models. You had better to explain in the text instead of table notes. I don't think uni-variate models are needed.

The text has been revised to note that walkability and physical activity were estimated simultaneously in one model. As previously noted, we have removed the unadjusted models from Tables 2 and 4 and the corresponding Results and Discussion sections.

L.331

The fact that walking for transportation and walking for recreation could not be distinguished should be written in limitation section. Long discussion is unnecessary here.

We have removed the reference to the limitations of the IPAQ categorical physical activity scores from this section. The limitation is discussed in Section 4.1 Strengths & Limitations.

L340

The discussion of material and social deprivation is not the main concern of your study. I think that a long discussion is unnecessary.

We have removed the section from the Discussion.

L.372

What do you mean by the term "mixed effects"? Usually refers to a statistical model containing both fixed effects and random effects.

With the removal of the data from the unadjusted model, we have subsequently deleted this statement from the text.

Reviewer 2 Report

This submission investigated the association between walkability and self-reported physical activity on chronic health conditions in an Atlantic Canadian population. I like to give the following comments.

  1. Definition of neighborhood walkability seems not introduced in clear.
  2. How to distinguish low-to-high walkability? Please introduce the merits of CAN-ALE Index in detail.
  3. The association between neighborhood walkability and physical activity was not observed in this report. Unable to have the walking for transportation or recreation needs the reason(s).
  4. History of the disease in participants was ignored. Why?
  5. A large sample of Nova Scotia participants needs to show in clear. How many participants joined in this study?
  6. Bias in self-report data did not discuss in clear.
  7. Conclusion seems more suitable to follow the limitations as the perspective. New conclusion shown the novel finding even the negative result may strengthen this report.

Author Response

Reviewer #2

Comments and Suggestions for Authors

This submission investigated the association between walkability and self-reported physical activity on chronic health conditions in an Atlantic Canadian population. I like to give the following comments.

  1. Definition of neighborhood walkability seems not introduced in clear.

We have revised the text to more clearly define neighborhood walkability (i.e., design features that can promote walking and access to walkable destinations). L54-55

  1. How to distinguish low-to-high walkability? Please introduce the merits of CAN-ALE Index in detail.

We have revised the methods to clearly indicate the merits of the CAN-ALE index.

  1. The association between neighborhood walkability and physical activity was not observed in this report. Unable to have the walking for transportation or recreation needs the reason(s).

It appears that the reviewer is asking why we were not able to distinguish between walking for recreation vs. walking for transportation? If we have interpreted the comment correctly, we would refer the reviewer to the limitations section where we discuss the limitations of using a categorical measure of physical activity (low, moderate, high) as determined by the IPAQ.  

  1. History of the disease in participants was ignored. Why?

While we are not certain what “history” the reviewer is referring to, we have assumed that the review is referencing “family history” (as participant history has been summarized as multi-morbidity). Again, if we have interpreted the comment correctly, given the incompleteness of the family history data, we elected to not include this variable/risk factor in our final analyses.

  1. A large sample of Nova Scotia participants needs to show in clear. How many participants joined in this study?

We have revised Section 2.1 Design, to show that 31,173 participants from the four Atlantic provinces completed the baseline questionnaire. L77

  1. Bias in self-report data did not discuss in clear.

 We revised the study limitations to specify that self-reported data may be influenced by social bias and/or personal knowledge or willingness to report. L352-353

  1. Conclusion seems more suitable to follow the limitations as the perspective. New conclusion shown the novel finding even the negative result may strengthen this report.

We have revised the concluding statement to highlight the lack of association between walkability and chronic disease incidence. We further suggest that highly walkable neighborhoods may confer health benefit for those residents willing and able to engage in the activity.